# A Narrative Review on Non-Invasive Diagnostic Tools for the Analysis of Dental Arches in Orofacial Cleft Patients

**DOI:** 10.3390/children9101533

**Published:** 2022-10-07

**Authors:** Paula Karine Jorge, Eloá Cristina Passucci Ambrosio, Maria Aparecida de Andrade Moreira Machado, Thaís Marchini Oliveira, Ana Lúcia Pompeia Fraga de Almeida, Simone Soares

**Affiliations:** 1Department of Pediatric Dentistry, Orthodontics and Public Health, Bauru School of Dentistry, University of São Paulo, Alameda Dr. Octávio Pinheiro Brisolla 9-75, Bauru 17012-901, SP, Brazil; 2Hospital for Rehabilitation of Craniofacial Anomalies, University of São Paulo, Rua Sílvio Marchione 3-20, Bauru 17012-900, SP, Brazil; 3Department of Prosthodontics and Periodontology, Bauru School of Dentistry, University of São Paulo, Alameda Dr. Octávio Pinheiro Brisolla 9-75, Bauru 17012-901, SP, Brazil

**Keywords:** cleft lip, cleft palate, imaging, three-dimensional, diagnosis, dental arches

## Abstract

Background: It is necessary to analyze and monitor the facial growth of orofacial cleft patients. The documentation should therefore begin before and after primary surgeries. Technological evolution has transformed plaster models into 3D images through scanners that allow rational storage, manipulation, and rotation without the possibility of breakage or damage. Based on this fact, this narrative review aims to provide a feature on the three-dimensional tools available for the assessment of dental arches in children with orofacial cleft and mixed dentition. Material and Methods: Three databases were chosen (PubMed, ScienceDirect, and Scopus) and keywords were used to select papers. Results: During the database screening, 292 potentially relevant papers were found. After removing duplicates, titles, and abstracts, 32 papers presented qualifications for analysis. Through evaluating each document by reading it one by one, 24 papers fulfilled the eligibility criteria. Conclusions: It was concluded that digital tools—i.e., benchtop scanners which evaluate the dental arches of children with cleft lip, palate, and mixed dentition—are reproducible and reliable, without the use of ionizing radiation, allow storage, manipulation with sustainability, and help preserve the environment.

## 1. Introduction

Primary surgeries repair the anatomical defect but most often cause deleterious effects on facial growth, mainly related to the jaws [1,2,3,4,5,6]. 

As such, it is necessary to analyze and monitor the facial growth of individuals with cleft lip and palate. Facial growth documentation should begin before primary surgeries and continue after until five years of age. In addition to the documentation already included in the protocols and in plaster models, 3D photos can aid in the measurements and analyses of dental arches and facial growth.

The literature describes intraoral photos for the purposes of analyzing occlusion indexes. Plaster models are the gold standard [7] and plaster model images have been analyzed with accuracy [8]. Both intraoral photos and plaster models have proved to be reliable and reproducible [9]. Technological evolution has changed plaster models into 3D images through the use of scanners [10] that allow rational storage, manipulation, and rotation without the possibility of breakage or damage. With the use of software to carry out the evaluations, instead of using a caliper and rulers [11], more accurate linear [12] and angular measurements are obtained. In addition to these, more accurate measurements of area [6], volume [3], superimpositions [13], and occlusal contacts [14] are also obtained. All of these help to better understand what happens with the growing dental arches of patients undergoing the rehabilitation process. The software’s ability can be sufficiently precise and accurate enough to assess linear, angular, and volumetric measures, as well as surface areas and superimposition procedures [11].

We can highlight stereophotogrammetry as an aid in understanding how the facial growth and development of these patients occurs through 3D photos of the face [15,16,17], using computer programs that have linear, area, volume, and superimposition measurement tools.

Technology has become an ally in the study and observation of the craniofacial development and growth of patients with cleft lip and palate before, during, and after the rehabilitation process so that, with coherence and scientific evidence, we can improve treatment protocols. Thus, this narrative review aims to provide information on the three-dimensional tools available for the assessment of dental arches in children with cleft lip and palate at mixed dentition.

## 2. Materials and Methods

### 2.1. Search Strategy

PubMed, Scopus, and ScienceDirect were chosen as the databases reviewed. Additionally, the narrative review included papers only in the English language. The following keywords were used: Children; Cleft Lip; Cleft Palate; Imaging, Three-dimensional; and Dental Arches.

### 2.2. Inclusion Criteria

All studies that presented quantitative assessments, such as research, multicenter studies, randomized clinical trials, and retrospective clinical studies, were included. 

### 2.3. Selected Sample

-Maxillary dental arches of cleft lip and palate patients aged up to 12 years; -Optical devices, scanners, and stereophotogrammetry in order to reproduce 3D maxillary dental model;-Types of intervention, linear, angular, surface (area), volume measurements, and qualitative analysis of the occlusal index. Types of analysis of results, reliability, precision, repeatability (conventional vs. digital analysis), cross-sectional, and longitudinal analyses.

### 2.4. Exclusion Criteria

-Editorials, technical notes, opinion letters, case reports, case series, systematic reviews, and congress abstracts;-Mandibular dental arches;-Adolescents and adults;-Syndromes or other craniofacial anomalies;-Magnetic resonance imaging (MRI), computed tomography (CT), cone beam computed tomography (CBCT), ultrasound, radiographs, and photographs;-Quantitative or qualitative analysis of the face;-Impacted permanent teeth, secondary bone graft surgeries, and distraction osteogenesis;-Upper airways, and/or speech–language pathology assessment.

### 2.5. Study Selection

According to the inclusion and exclusion criteria, two examiners independently analyzed the titles and abstracts of the articles initially selected. The full texts were read whenever the title and abstract lacked sufficient information. This procedure avoided the exclusion of relevant papers. In the absence of consensus among the examiners considering the eligibility of some documents, a third reviewer participated in the scientific discussion.

### 2.6. Data Extraction

The examiners collected the following information after reading the full text of each paper: title, authors, year, and device were used to acquire the 3D image. Parameters were evaluated in the dental arches, anthropometric analysis software, selected sample, and type of study (either cross-sectional or longitudinal). All data collected were stored in a table (Microsoft Word 2019, Microsoft Corporation, Redmond, DC, USA). Figure 1 presents a flowchart of the paper selection process.

## 3. Results

During the database screening, 292 potentially relevant papers were found. After removing duplicates and reading the titles and abstracts, 32 papers were selected for analysis. Eight papers were excluded after carefully reading of the text. Twenty-four scientific articles were selected from between 2007 and 2022. All the studies evaluated were of participants with cleft lip and palate, 23 evaluated a UCLP patient and the other BCLP. Twelve studies were longitudinal, and the other twelve were cross-sectional (Table 1). Twenty-three studies used a scanner to obtain three-dimensional virtual dental arches, and the other used stereophotogrammetry equipment. The 3Shape Orthodontic Scanner (Copenhagen, Denmark) was the most used model (14 articles, as shown in Table 2). Fourteen different types of software were used in the studies. Mirror imaging software (Canfield Scientific Inc., Parsippany, NJ, USA) was the most used computer program (used in 5 articles). Linear measures were the most quantified (14 articles), while project palatal curve and superimposition were the least evaluated (1 article for each parameter, as shown in Table 3). Six of the selected articles were included in the reproducibility analysis (5 articles: occlusal index and 1 article: palatal surface area). Among these, one evaluated the accuracy (parameter assessed: area), while another evaluated the validity (parameter assessed: occlusal index, as shown in Table 4).

## 4. Discussion

In the last decade, technology and innovation have also assumed a prominent position in dentistry by providing researchers with more accurate measurements in growth analysis and dental arch evaluation. The study of orofacial development and the growth of patients with cleft lip and palate is widely evaluated before and after primary surgeries [1,2,3,4,5,6,18,19,20,21,22,25,30,31,32] and for the follow-up of specific therapies [1,24]. This orofacial growth and development evaluation aims at better techniques and surgical time due to the fact that gold standard surgical protocols have not yet been described. 

The image acquisition can be obtained from benchtop scanning to taking pictures. Bench scanners are the most used because they have certified technology with an affordable price. This type of equipment aims to digitize impressions, or dental models, in order to obtain 3D images, provide storage, manipulation, and the exchange in information between research centers for the purposes of cross-sectional and/or longitudinal studies as well as clinical follow-ups. However, non-dental scanners have been used as digitizers [26,27,28,29].

Another way of obtaining 3D images is through photographs, using devices such as stereophotogrammetry (Breuckmann SmartScan and Artec Eva) [23,28,29], which have the same functionality as scanners. After scanning, the images are analyzed by software that has tools capable of measuring linear distances [1,2,4,5,18,20,24,25,26,27,28,29,30,32,33,34], area [6,20,23,24,25,34], volume [3,25,29], occlusal index [19,21,22,31,33,35], angle [28,33,34] projection of palatal curve [24,34], and reproducibility [21,22,23,31,33,35]. Among the selected studies, linear measurements were the majority. The linear measurements promote the follow-up and evaluation of the anteroposterior and transversal growth of the maxilla, allowing the visualization of the malocclusion types [25,26] and arch shape [1].

The software can capture measurements of different magnitudes, including the analysis between three points (angles), between two points (area), and also three planes (volume). The analysis of the area measurements reveal the maxillary segments’ size, the arch’s total development, and the potential palate growth [23]. Volume is a broader measurement, considering the whole maxilla from the palate to the ridge and in covering all teeth. The volume is assessed from image superimposition, a relevant tool in the evaluation of craniofacial development, bone deficiency in the cleft region, and in monitoring the effect of rehabilitation protocols in patients with cleft lip and palate [3]. This technology, either intraoral or model scanning, proved to be a minimally invasive method without the use of ionizing radiation [36].

The presented technologies proved to be reliable and reproducible [21,22,23,31,33,35] for analyzing the effects of primary surgeries on dental arches [1,2,3,5,6,18,20,24,25,26,27,32], nasoalveolar devices [1,29], and the intermaxillary relationship [4,19,21,22,33,34] when comparing individuals with and without cleft lip and palate [4,28,30]. In the present study, six articles performed an analysis of accuracy, validity, and/or reliability, which corresponds to 25% of the selected articles (Table 4). All hardware and software applied in three-dimensional analysis must be tested before use in clinical cases (i.e., for diagnosis, planning, and clinical procedures) and in scientific studies. These are important criteria to guarantee the reliability of the sample, which will be evaluated in the virtual environment [37].

## 5. Conclusions

Based on the eligible studies of this narrative review, it is concluded that using digital tools and benchtop scanners in order to evaluate the dental arches of children with cleft lip and palate, at a mixed dentition, are reproducible, reliable, possible without the use of ionizing radiation, capable of allowing storage, allow manipulation with sustainability, and are able to assist with environment preservation.

## Figures and Tables

**Figure 1 children-09-01533-f001:**
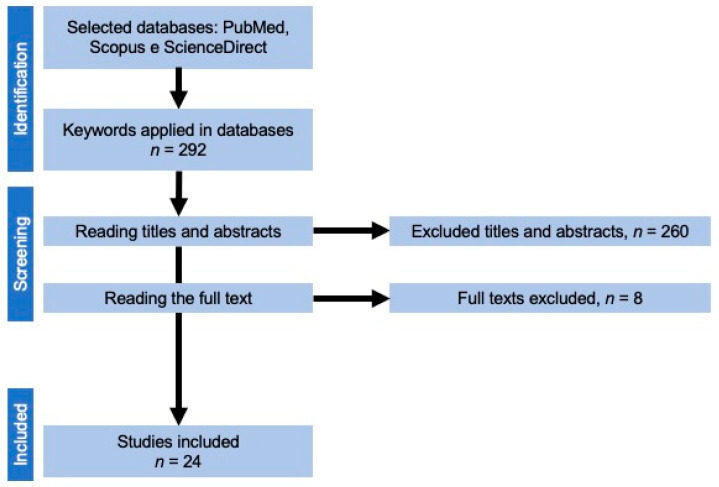
Flowchart of paper selection process.

**Table 1 children-09-01533-t001:** Studies selected for the narrative review.

	Title	Author	Image Acquisition	Software	Parameters	Age	Sample Size	Study Type
1	Post-surgical effects on the maxillary segments of children with oral clefts: New three-dimensional anthropometric analysis	Ambrosio et al., 2018a [2]	3Shape R700™ Orthodontic Scanner (Copenhagen, Denmark)	Mirror imaging software, Canfield Scientific Inc.	IC; IT; IC’; and IT’	3, 12, and 24 months	Unilateral Cleft Lip and Palate (UCLP)—30 Unilateral Cleft Lip and Alveolus (UCLA)—30	Longitudinal
2	Longitudinal morphometric analysis of dental arch of children with cleft lip and palate: 3D stereophotogrammetry study	Ambrosio et al., 2018b [18]	3Shape R700™ Orthodontic Scanner (Copenhagen, Denmark)	Mirror imaging software, Canfield Scientific Inc.	CC’; TT’; I-CC’; I-TT’; and area	3, 12, and 24 months	UCLP—30 Complete Unilateral Cleft Lip (UCL)—30	Longitudinal
3	Digital Volumetric Monitoring of Palate Growth in Children With Cleft Lip and Palate	Ambrosio et al., 2022 [3]	3Shape R700™ Orthodontic Scanner (Copenhagen, Denmark)	Mirror imaging software, Canfield Scientific Inc.	Volume and maxillary arch	3, 12, and 24 months	UCLP—20 UCL—21	Longitudinal
4	Dental Arch Relationships on Three-Dimensional Digital Study Models and Conventional Plaster Study Models for Patients with Unilateral Cleft Lip and Palate	Asquith et al., 2012 [19]	Orthodontic Study Model Scanner (3Shape A/S, Copenhagen, Denmark)	OrthoAnalyzer^TM^ software, 3Shape, Copenhagen, Denmark	Occlusion Huddart Bodeham index	5 years	UCLP—30	Transversal
5	The effect of lip closure on palatal growth in patients with unilateral clefts	Bruggink et al., 2020 [20]	3Shape R500 3D Dental Laser scanner (3Shape R, Copenhagen, Denmark).	(MATLAB R 2018b, The Mathworks, Inc., Natick, MA, USA).	TT; CC; A(I)-CC; A(I)-TT; SS; and area	4–8; 12 months	Control group (without cleft)—70 UCLP—28	Longitudinal
6	The 5-year-old ‘Index: determining the optimal format for rating dental arch relationships in unilateral cleft lip and palate	Chawla et al., 2012 [21]	R640 3Shape Desktop study model scanner (3Shape A/S, Copenhagen, Denmark).	3Shape viewing software (3Shape A/S).	ATTACK Index – photo; 3D dental cast images; and reproducibility	5 years	UCLP—45	Transversal
7	Three-Dimensional Digital Models for Rating Dental Arch Relationships in Unilateral Cleft Lip and Palate	Chawla et al., 2013 [22]	R640 3Shape Desktop study model scanner (3Shape A/S, Copenhagen, Denmark).	3Shape viewing software (3Shape A/S).	ATTACK Index—photo; 3D dental cast image; and reproducibility	5 years	UCLP—45	Transversal
8	Evaluation of a Three-Dimensional Stereophotogrammetric Method to Identify and Measure the Palatal Surface Area in Children with Unilateral Cleft Lip and Palate	de Menezes et al., 2016 [23]	VECTRA-3D, (Canfield Scientific Inc., Fairfield, NJ, USA)	Mirror imaging software, Canfield Scientific Inc.	Area and reproducibility	10 days to 1 year	UCLP—32	Longitudinal
9	Growth of Palate in Unilateral Cleft Lip and Palate Patients Undergoing Two-stage Palatoplasty and Orthodontic Treatment	Eriguchi et al., 2018 [24]	Scanner (Matsuo Sangyo Co., Tokyo, Japan)	CAD software Surface (Image ware, Tokyo, Japan).	CC’; EE’; MM’; TT’; project palatal curve; and area	8 to 16 years	UCLP—20	Longitudinal
10	Three-dimensional evaluation of the maxillary arch and palate in unilateral cleft lip and palate subjects using digital dental casts	Generali et al., 2017 [25]	3Shape R700™ Orthodontic Scanner (Copenhagen, Denmark)	Rapidform^TM^ 2006 (INUS Technology, Tokyo, Japan).	CC’; MM’; area; and volume	5 to 11 years	Control group (without cleft)—19 UCLP—19	Transversal
11	Effects of Multiple Factors on Treatment Outcome in the Three-Dimensional Maxillary Arch Morphometry of Children with Unilateral Cleft Lip and Palate	Haque et al., 2020 [26]	Next Engine laser scanner (Santa Monica, CA, USA).	Mimics software (Leuven, Belgium).	CC’; MM’; and I-MM’	7 years	UCLP—85	Transversal
12	An Investigation of Three-Dimensional Maxillary Arch Morphometry of Children with Unilateral Cleft Lip and Palate	Haque et al., 2021 [27]	Next Engine laser scanner (Santa Monica, CA, USA).	Mimics software (Leuven, Belgium).	CC’; MM’; and I-MM’	7 years	UCLP—85	Transversal
13	Three-dimensional development of the upper dental arch in unilateral cleft lip and palate patients after early neonatal cheiloplasty	Hooffmanova et al., 2018 [28]	Breuckmann SmartScan scanner (Aicon 3D Systems GmbH, Braunschweig, Germany)	RapidForm XOS software (INUS Technology, Inc., Seoul, Korea)	SS’; C’T’; MM’; CC’distal; CT; TT’; I-TT’; ScS’; and S’CmesialC’distal (angle) superimposition	3 days to 10 months	UCLP—36 incomplete UCLP—20	Longitudinal
14	Comparison of two treatment protocols in children with unilateral complete cleft lip and palate: Tridimensional evaluation of the maxillary dental arch	Jorge et al., 2016 [1]	3Shape R700™ Orthodontic Scanner (Copenhagen, Denmark)	OrthoAnalyzer^TM^ software, 3Shape	CC; TT; PY; PP(SS’); UU; and Ii	3 to 18 months	UCLP Hotz 24 UCLP HRAC 23	Longitudinal
15	Three-dimensional evaluation of the effect of nasoalveolar molding on the volume of the alveolar gap in unilateral clefts	Lautner et al., 2020 [29]	Artec Eva 3D scanner (Artec3D, Luxembourg)	Geomagic Control software version 9 (3D Systems Corporation, Rock Hill, SC, USA).	Volume and SS’	1 day to 4 months	UCLP NAM 10 UCLP without NAM—10	Longitudinal
16	Evaluation of the intercanine distance in newborns with cleft lip and palate using 3D digital casts	Mello et al., 2013 [30]	3Shape R700™ Orthodontic Scanner (Copenhagen, Denmark)	OrthoAnalyzer^TM^ software, 3Shape	CC’	3 to 9 months	Without cleft—19 UCLP—50 BCLP—25	Transversal
17	Analysis of Dental Arch in Children with Oral Cleft Before and After the Primary Surgeries	Mello et al., 2019 [5]	3Shape R700™ Orthodontic Scanner (Copenhagen, Denmark)	3D Software Appliance Designer, 3Shape	CC’; TT’; I-TT’; and I-CC’	3 to 24 months	UCLP—36 UCL—33 CP—30	Longitudinal
18	Evaluation of cheiloplasty and palatoplasty on palate surface area in children with oral clefts: longitudinal study	Prado et al., 2021 [6]	3Shape R700™ Orthodontic Scanner (Copenhagen, Denmark)	Mirror imaging software, Canfield Scientific Inc.	Area	3 months to 5 years	UCL - 18 UCLP - 33 CP - 10	Longitudinal
19	Anthropometric Analysis of the Dental Arches of Five- Year-Old Children with Cleft Lip and Palate	Rando et al., 2018 [4]	3Shape R700™ Orthodontic Scanner (Copenhagen, Denmark)	3D Software Appliance Designer, 3Shape	CC’; MM’; Maxilla; and mandible	5 years	Control—30 UCL—30 UCLP—30 CP—30	Transversal
20	Rating dental arch relationships and palatal morphology with the EUROCRAN index on three different formats of dental casts in children with unilateral cleft lip and palate	Sabelis et al, 2016 [31]	Orthoproof, Doorn, The Netherlands	Digimodel^®^ (Ortholab BV, Doorn, The Netherlands)	Eurocran Index; photo; dental cast; and reproducibility	9 years	UCLP—45	Transversal
21	3D analysis of effects of primary surgeries in cleft lip/palate children during the first two years of life	Sakoda et al., 2017 [32]	3Shape R700™ Orthodontic Scanner (Copenhagen, Denmark)	OrthoAnalyzer^TM^ software, 3Shape	CC’; TT’; I-TT’; and I-CC	3 months to 2 years	UCLP—25 CP—29	Longitudinal
22	Retrospective Evaluation of Treatment Outcome in Japanese Children With Complete Unilateral Cleft Lip and Palate. Part 1: Five-Year-Old’ Index for Dental Arch Relationships	Suzuki et al., 2007 [33]	Vivid-700 laser scanner (Minolta Co., Osaka, Japan).	Software Vivid-700 laser scanner (Minolta Co., Osaka, Japan).	Huddart Bodeham occlusion; index; ATTACK index; CC’; MM’; and reproducibility comparing caliper and 3D image	4 to 6 years	UCLP—136	Transversal
23	Orthodontic characteristics of maxillary arch deficiency in 5-year-old patients undergoing unilateral cleft lip and palate repair with and without early gingivoplasty	Wojtaszek-Slominska et al., 2010 [34]	3-D Picza 4 scanner (Roland DG Corporation, Model PIX-4, Shizuoka-ken, Japan)	Z dimension to visualize them and perform metric analysis (computer pro- gram Ortbaz-R, Medical University, Gdansk, Poland)	CC’; TT’; EE (MM’); ITT’; CTT’; and C’T’T.	4 to 6 years	UCLP—120	Transversal
24	A Comparison of Three Viewing Media for Assessing Dental Arch Relationships in Patients with Unilateral Cleft Lip and Palate	Zhu et al., 2016 [35]	Scanner (Lythos^TM^ Digital Impression System, Ormco, Glendora, CA, USA)	VRMesh Design (Version 5.0, VirtualGrid, Bellevue, DC, USA)	GOSLOW index; and Reproducibility	9 years	UCLP—29	Transversal

**Table 2 children-09-01533-t002:** Devices for dental arch digitalization.

Scanner Devices	Brands and Models	Number of Papers
Bench Scanner	3Shape Orthodontic Scanner (Copenhagen, Denmark)	14
Bench Scanner *	Next Engine laser scanner (Santa Monica, CA, USA)	2
Stereophotogrammetry	VECTRA-3D, (Canfield Scientific Inc., Fairfield, NJ, USA)	1
Bench Scanner	Scanner (Matsuo Sangyo Co., Tokyo, Japan)	1
Scanner—Photo 3D *	Breuckmann SmartScan scanner (Aicon 3D Systems GmbH, Braunschweig, Germany)	1
Manual Scanner—Photo 3D *	Artec Eva 3D scanner (Artec3D, Luxembourg)	1
Bench Scanner	Orthoproof, Doorn, The Netherlands	1
Bench Scanner	Vivid-700 laser scanner (Minolta Co., Osaka, Japan).	1
Bench Scanner	3-D Picza 4 scanner (Roland DG Corporation, Model PIX-4, Shizuoka-ken, Japan)	1
Scanner manual	Scanner (Lythos^TM^ Digital Impression System, Ormco, Glendora, CA, USA)	1

* This is not a dental scanner.

**Table 3 children-09-01533-t003:** Software and measures used in the selected studies.

Software	Measurements	Number of Papers
Mirror imaging software (Canfield Scientific Inc., Fairfield, CT, USA)	Linear Area Volume	5
OrthoAnalyzer^TM^ software, (3Shape)	Occlusal index	4
3Shape viewing software (3Shape)	Occlusal index	2
3D Software Appliance Designer (3Shape)	Linear	2
Mimics software (Belgium).	Linear	2
(MATLABR 2018b, The Mathworks, Inc., Natick, MA, USA).	Linear Area	1
CAD software Surface (Image ware, Tokyo, Japan).	Linear Area Project palatal curve	1
Rapidform^TM^ 2006 (INUS Technology, Tokyo, Japan).	Linear Area Volume Angle Superimposition	1
RapidForm XOS software (INUS Technology, Inc., Seoul, Korea)	Linear	1
Geomagic Control software version 9 (3D Systems Corporation, Rock Hill, SC, USA).	Linear Volume	1
Digimodel^®^ (Ortholab BV, Doorn, The Netherlands)	Occlusal index	1
Software Vivid-700 laser scanner (Minolta Co., Osaka, Japan).	Linear Occlusal index	1
Z dimension to visualize them and perform metric analysis (computer program Ortbaz-R, Medical University, Gdansk, Poland)	Linear Angle	1
VRMesh Design (Version 5.0, VirtualGrid, Bellevue, DC, USA)	Occlusal index	1

**Table 4 children-09-01533-t004:** Accuracy, validity, and reproducibility of the diagnostic tools.

Title	Author	Hardware	Software	Accuracy	Validity	Reproducibility
The 5-year-old ‘Index: determining the optimal format for rating dental arch relationships in unilateral cleft lip and palate	Chawla et al., 2012 [21]	R640 3Shape Desktop study model scanner (3Shape A/S, Copenhagen, Denmark).	3Shape viewing software (3Shape A/S)	There is no information	There is no information	Weighted kappa values (0.68 to 0.91)
Three-Dimensional Digital Models for Rating Dental Arch Relationships in Unilateral Cleft Lip and Palate	Chawla et al., 2013 [22]	R640 3Shape Desktop study model scanner (3Shape A/S, Copenhagen, Denmark).	3Shape viewing software (3Shape A/S)	There is no information	Weighted kappa values (0.69 to 0.74)	Weighted kappa values (0.74 to 0.87)
Evaluation of a Three-Dimensional Stereophotogrammetric Method to Identify and Measure the Palatal Surface Area in Children with Unilateral Cleft Lip and Palate	de Menezes et al., 2016 [23]	VECTRA-3D, (Canfield Scientific Inc., Fairfield, CT, USA)	Mirror imaging software, Canfield Scientific Inc.	Paired Student’s t tests. Valor de p entre 0.077 a 0.622	There is no information	Paired Student’s t tests *p* value ranging from 0.81 to 0.92
Rating dental arch relationships and palatal morphology with the EUROCRAN index on three different formats of dental casts in children with unilateral cleft lip and palate	Sabelis et al, 2016 [31]	Orthoproof, Doorn, The Netherlands	Digimodel^®^ (Ortholab BV, Doorn, The Netherlands)	There is no information	There is no information	Intra-class correlation coefficient (0.258 to 0.866)
A Comparison of Three Viewing Media for Assessing Dental Arch Relationships in Patients with Unilateral Cleft Lip and Palate	Zhu et al., 2016 [35]	Scanner (Lythos^TM^ Digital Impression System, Ormco, Glendora, CA, USA)	VRMesh Design (Version 5.0, VirtualGrid, Bellevue, DC, USA)	There is no information	There is no information	Weighted kappa values (0.63 to 0.88)
Retrospective Evaluation of Treatment Outcome in Japanese Children With Complete Unilateral Cleft Lip and Palate.	Suzuki et al., 2007 [33]	Vivid-700 laser scanner (Minolta Co., Osaka, Japan)	Software Vivid-700 laser scanner (Minolta Co., Osaka, Japan)	There is no information	There is no information	Weighted kappa value (0.611)

## Data Availability

Not applicable.

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
