# Peer review of "A Narrative Review on Non-Invasive Diagnostic Tools for the Analysis of Dental Arches in Orofacial Cleft Patients"

_children, 2022, doi:10.3390/children9101533_

Round 1

Reviewer 1 Report

Thanks for this interesting paper about the different tools used to assess dental arches in patients with CLP.

I have the following points:

Title

1- It should be clearly stated that the reviewed tools are not radiation-based. In other words, the review concentrates on non-invasive methods of studying dental arches. This should be given clearly in the title.

Abstract

2- The abstract should be structured to make its readability better. Please consider adding the following subheadings: Background, Materials and Methods, Results, Conclusions. Please read the Instruction to the Authors to adhere to the given guidelines.

Introduction

3- The first paragraph includes 17 references that are not directly related to the topic of this review. Please consider reducing the references from 17 to only 8 to 10 citations.

4- Regarding the third paragraph of the Introduction, Page 1 Line 43, you say that the "plaster models proved to be more reliable than photos". This should be modified to reflect the different opinions about this conclusion. Several intra-oral calibrated images assess dental arches with good reproducibility and accuracy. Please modify and depend on this citation:

Alrasheed WA, Owayda AM, Hajeer MY, Khattab TZ, Almahdi WH. Validity and Reliability of Intraoral and Plaster Models' Photographs in the Assessment of Little's Irregularity Index, Tooth Size-Arch Length Discrepancy, and Bolton's Analysis. Cureus. 2022 Mar 11;14(3):e23067. doi: 10.7759/cureus.23067. PMID: 35308184; PMCID: PMC8920827.

5 - Regarding using software to do different measurements on dental arches with high reproducibility (Page 2, Line 49), please consider adding the relevant paper to this sentence.

Al-Rayes NZ, Hajeer MY. Evaluation of occlusal contacts among different groups of malocclusion using 3D digital models. J Contemp Dent Pract. 2014 Jan 1;15(1):46-55. doi: 10.5005/jp-journals-10024-1486. PMID: 24939264.

6 - Regarding the ability of the programs to measure linear and angular distances, produce volumetric and surface areas assessments and do some superimpositional procedures in stereophotogrammetry.

Materials and Methods

7 - Please delete the URLs of the given databases.

8 - Please add the Web of Science database to your search strategy.

9 - Where is the Table that shows your search strategy?

10 - Please explain the reason for excluding MRI, CT, CBCT, and ultrasonography from your review.

Results

11- A table is required to show the accuracy, validity, and reproducibility of the included diagnostic tools. Please add this table from the collected papers. This table should take into account the hardware and software used.

Discussion

12 - Please add a paragraph in your discussion about the accuracy, validity, and reliability of the included diagnostic tools based on the requested table in the Results section (above).

Author Response

September 27th, 2022.

Dear Editor,

            Enclosed is the revised manuscript children-1899261, entitled “A narrative review on non-invasive diagnostic tools for the analysis of dental arches in orofacial cleft patients” that we would like to be considered for publication in the Children.

Please find also a letter explaining, point-by-point, the changes made in response to the critiques that we received. Our answers are in red in response to review.

We thank you and the reviewers for the excellent suggestions given to us. We firmly believe that your comments and suggestions have improved significantly our manuscript.

Manuscript: Point-by-point review

The authors sincerely thank the Reviewers for their excellent suggestions that have certainly improved our manuscript.

Manuscript #children-1899261

Reviewer: 1

Title

1- It should be clearly stated that the reviewed tools are not radiation-based. In other words, the review concentrates on non-invasive methods of studying dental arches. This

should be given clearly in the title.

RESPONSE: Thank you for your suggestion. We add the words non-invasive in the title.

Abstract

2- The abstract should be structured to make its readability better. Please consider adding the following subheadings: Background, Materials and Methods, Results, Conclusions.

Please read the Instruction to the Authors to adhere to the given guidelines.

RESPONSE: We added in structured subheadings in the abstract.

Introduction

3- The first paragraph includes 17 references that are not directly related to the topic of this review. Please consider reducing the references from 17 to only 8 to 10 citations.

RESPONSE: We removed 11 citations as suggested.

4- Regarding the third paragraph of the Introduction, Page 1 Line 43, you say that the "plaster models proved to be more reliable than photos". This should be modified to

reflect the different opinions about this conclusion. Several intra-oral calibrated images assess dental arches with good reproducibility and accuracy. Please modify and depend on this citation: Alrasheed WA, Owayda AM, Hajeer MY, Khattab TZ, Almahdi WH. Validity and Reliability of Intraoral and Plaster Models' Photographs in the Assessment of Little's Irregularity Index, Tooth Size-Arch Length Discrepancy, and Bolton's Analysis. Cureus. 2022 Mar 11;14(3):e23067. doi: 10.7759/cureus.23067. PMID: 35308184; PMCID: PMC8920827.

RESPONSE: We changed the sentence as suggested and it can be seen as below:

“Plaster models are the gold standard (7) and plaster model images have been analyzed with accuracy (8). Both intraoral photos and plaster models proved to be reliable and reproductible (9)”.

5 - Regarding using software to do different measurements on dental arches with high reproducibility (Page 2, Line 49), please consider adding the relevant paper to this

sentence.

Al-Rayes NZ, Hajeer MY. Evaluation of occlusal contacts among different groups of malocclusion using 3D digital models. J Contemp Dent Pract. 2014 Jan 1;15(1):46-55. doi: 10.5005/jp-journals-10024-1486. PMID: 24939264.

RESPONSE: We added the paper and thanks for suggestion.

6 - Regarding the ability of the programs to measure linear and angular distances, produce volumetric and surface areas assessments and do some superimpositional procedures in stereophotogrammetry.

RESPONSE: We added the information in the text.

“The software’s ability can be sufficiently precise and accurate to assess linear, angular, volumetric measures, surface areas, and superimposition procedures (11).

7 - Please delete the URLs of the given databases.

RESPONSE: We deleted the URLs.

8 - Please add the Web of Science database to your search strategy.

RESPONSE: We thank for the suggestion but this is a narrative review and we can use in a future paper.

9 - Where is the Table that shows your search strategy?

RESPONSE: Figure 1, flowchart of paper selection process.

10 - Please explain the reason for excluding MRI, CT, CBCT, and ultrasonography from your review.

RESPONSE: The ionization exams were excluded because the main target was dental cast. The dental casts are a non-invasive procedure, and for the first stages of the growth, the radiography is not necessary exam to submit the children.

Results

11- A table is required to show the accuracy, validity, and reproducibility of the included diagnostic tools. Please add this table from the collected papers. This table should take

into account the hardware and software used.

RESPONSE: Thank you for your suggestion. The table was added (Table 4).

Discussion

12 - Please add a paragraph in your discussion about the accuracy, validity, and reliability of the included diagnostic tools based on the requested table in the Results section (above).

RESPONSE: Thank you for your suggestion, and we added a new paragraph in the discussion about this new issue.

In the present study, six articles performed analysis of accuracy, validity, and/or reliability, which corresponds to 25% of the selected articles (Table 4). All hardware and software applied in three-dimensional analysis must be tested before the use in clinical cases (for diagnosis, planning and clinical procedures) and in scientific studies. These are important criteria to guarantee the reliability of the sample, which will be evaluated in the virtual environment [37]."

Reviewer 2 Report

In the manuscript titled ‘A narrative review on diagnostic tools for the analysis of dental arches in orofacial cleft patients’, authors Jorge et. al. have provided a narrative review of current 3d digital tools available for assessment of dentition and oral structures in individuals with orofacial cleft. The literature search and justification of the selected articles for analyses is thorough and the summary of results is commendable. Overall, the findings by authors will help the dental community and shed new light on the reliability of moving towards digital methods over conventional methods in dental and oral sciences. However, there are some major concerns that need to be addresses as mentioned below

1.      The authors summarize the results in tables however, there is no attempt to provide a description of their literature analyses. I find the result section to be completely lacking in material. Each table and aspect of the review should be described in the results section, so that the data presented is better interpreted by the readers.

2.      The authors continue using the term ‘intraoral scanners’ in introduction (line 53) and in conclusion (line 173). However, the review did not include studies with intraoral scans and the authors also mention this in the discussion (line 137). Hence, I strongly recommend rewording the introduction and conclusion to avoid mentioning intraoral scans in this article.

Author Response

Dear Editor,

            Enclosed is the revised manuscript children-1899261, entitled “A narrative review on non-invasive diagnostic tools for the analysis of dental arches in orofacial cleft patients” that we would like to be considered for publication in the Children.

Please find also a letter explaining, point-by-point, the changes made in response to the critiques that we received. Our answers are in red in response to review.

We thank you and the reviewers for the excellent suggestions given to us. We firmly believe that your comments and suggestions have improved significantly our manuscript.

Reviewer: 2

  1. The authors summarize the results in tables however, there is no attempt to provide a description of their literature analyses. I find the result section to be completely lacking in material. Each table and aspect of the review should be described in the results section, so that the data presented is better interpreted by the readers.

RESPONSE: Thank you for your suggestion, and we added a new paragraph in the discussion about the tables.

“Twenty-four scientific articles were selected between 2007 and 2022. All studies evaluated participants with cleft lip and palate, 23 evaluated UCLP and one BCLP patient. Twelve studies are longitudinal, and 12 are cross-sectional (Table 1). Twenty-three studies used scanner to obtain three-dimensional virtual dental arches, and one used stereophotogrammetry equipment. 3Shape Orthodontic Scanner (Copenhagen, Denmark) was the most used model (14 articles) (Table 2). Fourteen different software were used in the studies. Mirror imaging software (Canfield Scientific Inc.) was the most used computer program (5 articles). Linear measures were the most quantified (14 articles), while project palatal curve and superimposition were the least evaluated (1 article for each parameter) (Table 3). Six selected articles performed the reproducibility analysis (5 articles: occlusal index. 1 article: palatal surface area). Among these, one evaluated the accuracy (parameter: area) while another performed the validity (parameter: occlusal index) (Table 4).”

  1. The authors continue using the term ‘intraoral scanners’ in introduction (line 53) and in conclusion (line 173). However, the review did not include studies with intraoral scans and the authors also mention this in the discussion (line 137). Hence, I strongly recommend rewording the introduction and conclusion to avoid mentioning intraoral scans in this article.

RESPONSE: Thank you for your suggestion. The intraoral scanners were removed throughout text.

Round 2

Reviewer 1 Report

Thanks for addressing all of my raised issues in my previous review of your manuscript.